# Practical Design Considerations for Compact Array-Fed Huygens’ Dielectric Lens Antennas

**DOI:** 10.3390/s19030538

**Published:** 2019-01-28

**Authors:** Bora Seo, Inseop Yoon, Jungsuek Oh

**Affiliations:** 1Department of Electronic Engineering, Inha University, Incheon 22212, Korea; seobora0520@gmail.com (B.S.); lunarmaestro@inha.edu (I.Y.); 2Institute of New Media and Communications and School of Electrical and Computer Engineering, Seoul National University, Seoul 08826, Korea

**Keywords:** dielectric lens, lens antennas, antenna arrays

## Abstract

This paper presents a practical design consideration for the dielectric lens based on Huygens’ principle (HP) at a short distance (=*λ*_0_/2) from a feed antenna to overcome the limitation of the conventional design method. It is suggested that certain ranges of dielectric thickness values are not considered to exclude undesired resonant effects that hamper the effectiveness of Huygens’ lens which relies on phase shifting elements. In the proposed HP-based design method, phase distributions are captured at the target distance away from the feed array for the two cases of 2 × 2 and 1 × 4 array antennas and based on these, the proposed lens topology is designed to compensate the phase distributions for gain enhancement. A case study shows that the proposed HP-based design approach considering the actual phase information and undesired dielectric resonant phenomenology can achieve a gain enhancement of up to 5.34 dB compared to the conventional dielectric lens, depending on the feed array arrangement that can render circular or elliptic shapes of phase distributions for radiated fields.

## 1. Introduction

Dielectric lens antennas are attracting attention from antenna engineers for millimetre wavelengths or shorter wavelength applications where they become compact, especially for configurations with integrated lens antennas [1]. Convex dielectric lenses can increase the radiating aperture and thus antenna gain by transforming spherical waves into planar waveforms [2]. Since such lenses are based on simple topologies in terms of design and fabrication and thus can be a reliable alternative to reflector antennas, there have been many preceding studies on the dielectric lens and metasurfaces [3,4,5,6,7]. The most widely used design methodologies for the convex dielectric lens employ the parabolic equation (PE) which assumes that the electromagnetic wave radiated by the feed antenna is an ideally spherical wave produced by a single point source [8]. Therefore, the phase distribution of radiated fields is considered to render circular and symmetrical shapes and thus designed lenses have an axially symmetrical shape. However, this assumption may result in performance degradation for different sources such as multi-source antennas arranged along an axis whose radiated fields render elliptic shapes of phase distribution.

In this paper, practical design considerations and an approach for an array-fed dielectric lens based on Huygens’ principle (HP) are introduced. Phase distributions of the radiated fields emitted from the antennas are captured for specific polarization over the lens aperture. Next, the proposed method is applied for these massive data to design the HP based lens [9]. The Huygens’ principle states that each locus of a wave excites the local matter which reradiates secondary wavelets and all wavelets superpose to a new, resulting wave (the envelope of those wavelets) and so on [10]. Thus, it is possible to design a lens for multi-source antennas whose radiated fields may render elliptical or asymmetric shapes of phase distribution. Accordingly, the shape of the lens can be designed not only symmetrically but also asymmetrically or arbitrarily. This suggests that the proposed HP method can overcome the limits in determining the geometry of the conventional PE method-based lens. Section 2 describes the conventional PE-based and the proposed HP-based design methods for the dielectric lens and compares them. Section 3 discusses a procedure for determining the appropriate thickness of the dielectric lens to utilize the phase shifting operation as well as presenting the corresponding simulation results. Finally, various features of the conventional PE lens and the proposed HP lens are compared and analysed.

## 2. Lens Design Method

### 2.1. PE Method

This section presents a design procedure of a conventional PE-based dielectric lens and related dimension parameters. The lens geometry for the PE method is shown in Figure 1. The PE method assumes that propagating rays are illuminated from a single point source, as shown in Figure 1.

In Figure 1, Snell’s law of refraction is applied at the bottom surface of the lens as follows:(1)$$\frac{\partial r\left(\eta \right)}{\partial \eta }=\frac{r\left(\eta \right)n\sin\left(\gamma -\eta \right)}{1-n\cos\left(\gamma -\eta \right)}$$
where *η* is the independent variable and *n* is a refractive index. Also, the following electrical path length condition can be imposed:(2)$$r+nl+s=F+nT_{\max}$$
where
(3)$$s=F+T_{\max}-r\cos(\eta )-l\cos(\gamma )$$
(4)F=rcos(η)
as discussed in Reference [1]. Similar to (1), Snell’s law of refraction is applied at the top surface of the lens as follows:(5)nsin(α)=sin(α+γ)
At the beginning of the design procedure, *F* and *T*_max_ are first chosen as input parameters. *F* is the distance between the lens and the antenna and *T*_max_ is the thickness of the thickest part of the dielectric lens. The bottom surface of the lens is selected to be planar as a conventional reference geometry and the upper surface is determined by the simultaneous equations above. The radius at the bottom of the lens, *R_b_* is determined by the total reflection condition and the corresponding incidence critical angle is given as follows:(6)$$\alpha_{c}=\sin^{-1}(1/n)$$
where *α**_c_* is the ray’s critical incidence angle measured with respect to the local normal vector of the lens, $\hat{n}$ [1]. It should be noted that once *F*, *T_max_* and *n* in Figure 1 are chosen maximum available radius of the lens is decided by the total reflection condition at the top surface of the lens. In more detail, once a design point on the lens is chosen by *η* to decide the dielectric thickness at that point, *F*, *T_max_*, *n* and *η* become known. Accordingly, other five variables *r*, *γ*, *l*, *α* and *s* can be solved by five Equations (1)–(5). As *α* goes over *α_c_*, reflections on the top surface of the lens goes into the total reflection condition and thus no solutions exist for the five variables related to the lateral size of the lens by *R_b_* = *r* * sin(*η_max_*). Therefore, if one want to further increase the lateral size of the dielectric lens, the fixed values for *F*, *T_max_* and *n* must be changed. For example, the increase of *T_max_* allows for the increase of the lateral dimension of the lens. Unlike the PE method assuming a point source, HP method that can consider multiple sources practically allows for design of the wider dielectric lens. This is why once *F*, *T_max_* and *n* are fixed for fair comparisons, the lateral dimensions of the two different method-based lenses can be different, suggesting practical advantages of HP method.

Since most antenna applications use antenna arrays due to their beam steering capability, the assumption of a single point source in the above PE method limits the achievement of optimal lens gain for the multiple and arbitrarily arranged radiating sources.

### 2.2. HP Method

HP-based lens design methodologies allow the actual phase distributions of the radiated fields illuminated from the feed array to be considered. The lens geometry used to realize the HP method with a 2 × 2 patch array is shown in Figure 2. The macro lens aperture was designed by determining the required dielectric thickness to compensate for the phases of Huygens’ sources in the capture plane of Figure 2. This collimates all of the rays propagating through the phase capture plane.

The first step of the proposed design procedure is to measure the phases of E-fields radiating from a single or multi-source antenna at phase capture plane in Figure 2 for a target polarization. The phase capture plane is at a distance of *F* away from the feed array and is a starting position of the bottom surface of the lens. *F* is an input parameter that indicates the distance between the lens and the feed array. All phases of the propagating wave ($\varphi_{1},\varphi_{2},\cdots ,\varphi_{k}$) must be transformed to have the same phase value when they reach the target plane in Figure 2. Accordingly, the following condition can be imposed:(7)$$\varphi_{1}-\frac{2\pi nT_{\max}}{\lambda_{0}}=\varphi_{k}-\frac{2\pi nl}{\lambda_{0}}-\frac{2\pi s}{\lambda_{0}}\;\;\;\;\;\;\left(k=1,2,3,\cdots \right)$$
where *T*_max_ is the input parameter indicating the maximum thickness of the dielectric lens, *λ*_0_ is the free space wavelength and *n* is the refraction index of the lens material. Since *s* = *T*_max_ – *l*, the required dielectric thickness *l* can be expressed as
(8)$$l=T_{\max}-\frac{\lambda_{0}\left(\varphi_{1}-\varphi_{k}\right)}{2\pi \left(n-1\right)}$$
The HP-based lens is designed by building up a dielectric as thick as *l* at each point and the width of the lens is determined once *T*_max_ is chosen.

The proposed lens design method reflects the actual phase distribution of multiple feed elements and accordingly, the lens can have any shape corresponding to the phase distribution. Since the conventional PE method assumes there is a point source, it is assumed that the radiation pattern and the phase distribution are also symmetric. Therefore, the shape of the lens can be only symmetrical with respect to the radiation axis. On the other hand, the HP-based design method is much more useful for supporting a variety of feed array arrangements because both symmetric and asymmetric structures of the lens can be designed for a broader range of beam steering antennas.

## 3. Design Condition for Validation

The dielectric lenses are known to be significantly affected by the thickness of the dielectric medium [11,12,13,14]. The dielectric thickness at which the maximum gain at boresight, *T*, is obtained can be calculated [12] as
(9)$$T=\frac{\lambda }{2}(\frac{\phi }{2\pi }+0.5)+N\frac{\lambda_{0}}{2}$$
where *ϕ* is the phase of the reflection coefficient of the dielectric medium and *N* is the order of the resonant mode and equals 0, 1, 2, … and so forth. Figure 3 shows the simulation model and the total gain of the dielectric medium illuminated by a 1 × 4 antenna array as a function of its thickness. The distance between the antennas and the cylindrical dielectric was chosen to be *λ*_0_/2, based on the fact that microwave impedance matching features are repeated every *λ*_0_/2 and the width of the cylindrical dielectric is wide enough to cover the antennas. A patch element in the antenna arrays was designed to resonate at a centre frequency of 28 GHz and the dielectric constant of the cylindrical dielectric medium was 2.08. The gain of the 1 × 4 antenna array was 12.04 dBi without the dielectric medium. In the 1 × 4 antenna array, the width, *P_w_* and length, *P_l_*, of the patch were both 2.7 mm and the radius of the cylindrical dielectric medium, *R*, was 20 mm, as shown in Figure 3a. The centre distance between patches of the antenna array, *d*, was *λ*_0_/2. The total gain of the 1 × 4 antenna array-fed dielectric medium was simulated by using Ansys HFSS in the range of 1 to 30 mm for the thickness of the cylindrical dielectric medium, *h*. Figure 3b shows that the gain varied periodically at every *λ*_0_/2, as can be predicted from Equation (9). However, the resonant condition at the specific thickness made it difficult to compare the PE method with the HP method because the lens gain is critically affected by the thickness as well as the surface shape of the lens. It was observed that the antenna gain of the 1 × 4 antenna array could be increased up to 16.04 dBi depending on the thickness of the dielectric medium. It should be noted that as the thickness is adjusted to approach the resonant condition, the total antenna gain may be drastically lowered with beam steering compared to an ordinary lens designed based on the phase shifting condition. In this sense, gain enhancement from phase shifting collimation by engineering the shape of the dielectric lens is prohibited at some thickness values of the cylindrical dielectric medium, causing the aforementioned resonant condition and thus, a loss in beam steering capability. Therefore, the appropriate thickness of the dielectric lens should be selected by avoiding thicknesses related to the undesired resonance. It is obvious that the selected thickness becomes the aforementioned *T*_max_ in Figure 1 and Figure 2. In this study, *T*_max_ was chosen to be 19 and 21 mm for the 1 × 4 antenna array. When the thicknesses of the cylindrical dielectric medium were 19 mm and 21 mm, the antenna gains were 7.39 dBi and 12.32 dBi, respectively. In one case of selecting 7.39 dBi, the antenna beam is most widely spread and thus, this case has the greatest capacity to improve the total gain by using the dielectric lens. In the other case of selecting 12.32 dBi, the gain value is similar to the antenna gain without any dielectric medium and thus, this case has a minimum effect from the resonance of the dielectric medium.

Similarly, a 2 × 2 antenna array was also simulated to compare the symmetric (2 × 2) and asymmetric (1 × 4) shapes of phase distributions between the cases as shown in Figure 4. It should be noted that the gain variations for 2 × 2 antenna array have different shapes compared with the gain variations for 1 × 4 antenna array due to the superposition of electromagnetic waves radiated by differently arranged multiple sources. The width and length of a patch element in the square 2 × 2 antenna array were both 3.5 mm and the radius of the cylindrical dielectric medium was 10 mm. The gain of the 2 × 2 antenna array was 12.04 dBi without the cylindrical dielectric medium. For the 2 × 2 antenna array, *T*_max_ was chosen to be 4 and 26 mm. When the thicknesses of the cylindrical dielectric were 4 mm and 26 mm, the simulated antenna gains were 13.88 dBi and 14.64 dBi, respectively. Similar to the case of 1 × 4 antenna array, The 4 mm and 26 mm are chosen for total gain of the cylindrical dielectric-medium combined antenna array to be similar to the gain of the antenna array without any dielectric medium and thus gain enhancement effectiveness by phase compensation of the lens can be realized. The reason why 1 or 2 mm is not selected is that too small *T_max_* limits in maximum available lateral dimension (*R_b_*) of the lens due to total reflection condition on the top surface of the dielectric lens as stated in Section 2.1.

## 4. Bench Marketing

### 4.1. 2 × 2 Patch Antenna Array

As described in Section 2.1, PE design method for the dielectric lens establishes a lens surface function by determining the *F* and *T*_max_, as shown in Figure 1. The distance between the lens and the antenna, *F*, was set to be *λ*_0_/2 and the maximum thickness of the lens, *T*_max_, was set to be the selected value, as described in Section 3. The selected *T*_max_ values for the 2 × 2 antenna array were 4 mm and 26 mm and two different lenses were designed for the two thicknesses. Figure 5a shows the simulation model of the dielectric lens designed based on the PE method for the 2 × 2 antenna array. Since the conventional method assumes a point source, it can be seen that the lens is symmetrical with respect to the *z*-axis direction. The dimension parameters of the lenses in Figure 5a are listed in Table 1. When *T*_max_ is 4 mm, the *R_b_* is 4.6 mm where *R_b_* is radius at the bottom of the lens. When *T*_max_ is 26 mm, the *R_b_* is 16 mm. In Figure 1, *R_b_* is given by *r* × sin(*η_max_*). In more detail, total reflection condition on the top surface of the lens, *α* = *α_c_*, gives *r* and *η_max_* by solving Equations (1)–(5). It should be noted that as *α* goes over *α_c_*, reflections on the top surface of the lens goes into the total reflection condition and thus no solutions exist for *R_b_*. This limits in maximum available lateral dimension of the lens for the fixed *F*, *T_max_* and *n*. Through the aforementioned design procedure, when *F*, *T_max_* and *n* are 5.3 mm, 4 mm and 1.44, *R_b_* is decided to be 4.6 mm.

In the proposed HP method, the lens is designed by capturing the phase distributions at a target distance away from the feed array. Also, the phase distribution is captured for a target polarization of the E-field radiated from the feed, as mentioned in Section 2.2. The phase distributions are acquired by a full-wave simulator, Ansys HFSS. Phase information of an e-field component corresponding to antenna polarization is captured along the lens aperture at the target distance away from the antenna array. Figure 6 shows the phase distributions at a distance of *λ*_0_/2 from the 2 × 2 antenna array when there is no dielectric lens above the feed array. Since antenna elements in the array are arranged in 2 × 2 dimensions, a symmetrical arrangement, the shape rendered by the phase distributions is circular and symmetrical. Figure 5b shows the lens designed based on the HP method for the 2 × 2 antenna array and the shape of the lens follows the shape of the phase distributions. The dimension parameters of the lenses in Figure 5b are listed in Table 2. *a* is a major axis in the bottom of the lens and *b* is a minor axis. When *T*_max_ is 4 mm, *a* is 8.6 mm and *b* is 7.7 mm. When *T*_max_ is 26 mm, *a* is 21.3 mm and *b* is 19.5 mm.

The simulated results of the 2 × 2 antenna array-fed dielectric lenses operating at 28 GHz are shown in Figure 7. Radiation patterns on the E-plane and the H-plane of for each *T*_max_ are shown by comparing the PE method-based lens (PE lens) with the HP method-based lens (HP lens). The gains of the lens antennas are given in Table 3. The gain of the 2 × 2 antenna array was 12.04 dBi without the lens. When *T*_max_ is 4 mm, the antenna with the PE lens has a gain of 13.07 dBi and the antenna with the HP lens has a gain of 13.90 dBi. When *T*_max_ is 26 mm, the antenna with the PE lens has a gain of 19.02 dBi and the antenna with the HP lens has a gain of 19.40 dBi. When the *T*_max_ values are 4 mm and 26 mm, the HP lens achieves 0.83 dB and 0.38 dB higher gain enhancements than the PE lens, respectively. Since the 2 × 2 antenna array does not deviate significantly from the assumption of a point source in PE method, the shapes of the PE lens and HP lens are not so different and thus there is no big difference in total antenna gain. However, it was observed that there is still a benefit for improving the gain by using the HP lens.

### 4.2. 1 × 4 Patch Antenna Array

Similarly, the selected *T*_max_ values for the 1 × 4 antenna array were 19 mm and 21 mm and two different lenses were designed for the two thicknesses. Figure 8a shows the simulation model of the dielectric lens based on the PE method for the 1 × 4 antenna array. As in the PE method, the shape of the lens for the 1 × 4 antenna array is still symmetrical with respect to the *z*-axis direction. It is obvious that the PE method, which assumes a single point source, cannot fully reflect the asymmetrical features of the 1 × 4 feed array. The dimension parameters of the 1 × 4 antenna array-fed PE lens in Figure 8a are listed in Table 4. When *T*_max_ is 19 mm or 21 mm, the *R_b_* is 12.7 mm or 13.6 mm, respectively.

Figure 8b shows the lens based on the HP method for the 1 × 4 antenna array and the shape of the lens corresponds to the shape drawn by the phase distributions. Figure 9 shows cross sections of the 1 × 4 antenna array-fed HP lens in the XZ plane and the YZ plane to help understand the 3D topology of the lens. The dimension parameters of the lenses in Figure 8b are listed in Table 5. When *T*_max_ is 19 mm, *a* is 22 mm and *b* is 12 mm. When *T*_max_ is 21 mm, *a* is 23 mm and *b* is 12.7 mm. Figure 10 shows the phase distributions captured to design the 1 × 4 antenna array-fed dielectric lens at a distance of *λ*_0_/2 from the feed array. It is observed that the phase distributions draw a long oval shape.

The simulated radiation patterns of the lenses designed for 1 × 4 antenna array operating at 28 GHz are shown in Figure 11. Radiation patterns on the E-plane and the H-plane are shown by comparing the PE lens with the HP lens. The gains of the lens antennas are given in Table 6. The gain of the 1 × 4 antenna array was 12.32 dBi without the lens. When *T*_max_ is 19 mm, the 1 × 4 antenna array-fed PE lens has a gain of 13.53 dBi and the 1 x 4 antenna array-fed HP lens has a gain of 17.97 dBi. When *T*_max_ is 21 mm, the antenna with the PE lens has a gain of 12.68 dBi and the antenna with the HP lens has a gain of 18.11 dBi. When *T*_max_ is 19 mm or 21 mm, the HP lens achieves a 4.14 or 5.34 dB higher gain enhancement than the PE lens, respectively. Since the 1 × 4 antenna array deviates significantly from the assumption of a single point source in the PE method, the levels of gain enhancement of the PE lens and HP lens are very different. The PE method assumes that the radiation pattern or the phase distribution of radiated fields from the feed array are symmetrical but for the 1 × 4 feed array, they are asymmetric, as shown in Figure 10. Finally, it was found that this discrepancy can be fixed by employing the HP lens, leading to significant gain enhancement.

Figure 12 and Figure 13 and Table 7, Table 8, Table 9 and Table 10 show the steered radiation patterns at the phase offsets of 0, 45 and 90 degrees for PE and HP lenses with the different array arrangement such as 2 × 2 and 1 × 4 and the different thicknesses of the dielectric lens. It is found that for 2 × 2 antenna array, such a symmetrical arrangement, both of PE and HP lenses have similar beam steering features. However, for 1 × 4 antenna array HP lenses shows higher gain than PE lenses at all of the phase offsets. It is observed that the PE lens has higher gains at steer angles, not at *θ* = 0 degree. This suggests that the PE method using theoretically ideal assumption of a point source is not relevant for practical dielectric lens antennas that must consider many different arrangement cases for the feed antenna array.

## 5. Conclusions

A novel practical design approach for the dielectric lens based on Huygens’ principle at a short-distance (=*λ*_0_/2) away from the feed array was presented. It was demonstrated that employment of the lens designed based on the proposed practical considerations enables a gain enhancement of up to 5.34 dB for the 28 GHz antenna array whose radiated fields render elliptic shapes of phase distributions. The proposed design method that can consider actual field profiles radiated from an arbitrary source arrangement and the undesired resonant conditions of the dielectric medium overcome the limitations of conventional design methods based on the parabolic equation. It is expected that the proposed approach will be very useful for many different integration scenarios of dielectric lenses, which is essentially required for recent industrial systems such as repeaters and base stations in massive MIMO and 5G. In addition, recently evolving 3D/4D printing technology will allow for fabrication of smoothly varying HP dielectric lenses with high-resolution in the near future.

## Figures and Tables

**Figure 1 sensors-19-00538-f001:**
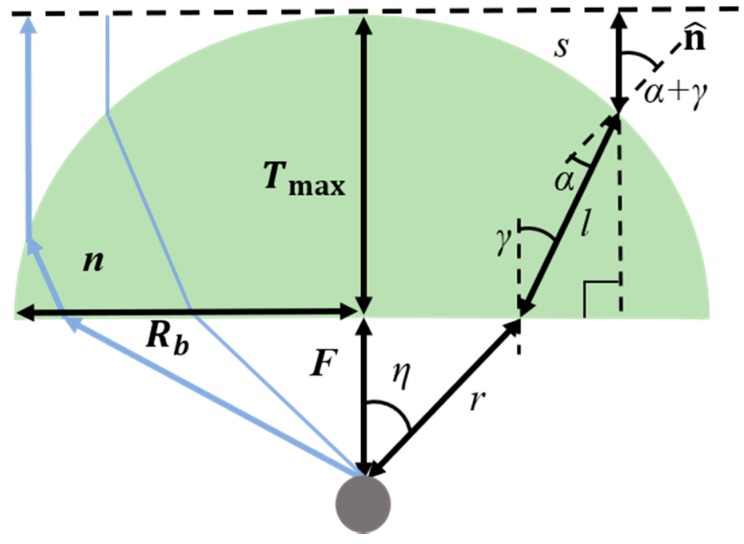
Geometry and dimension parameters of the dielectric lens based on the parabolic equation (PE) design method.

**Figure 2 sensors-19-00538-f002:**
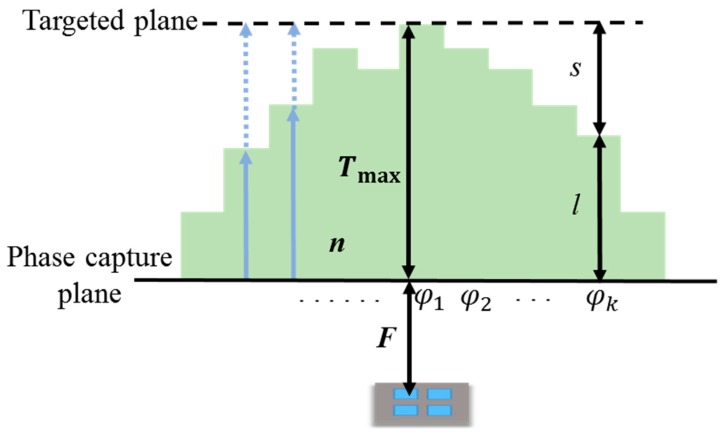
Geometry and dimension parameters of the dielectric lens based on the Huygens’ principle (HP) design method.

**Figure 3 sensors-19-00538-f003:**
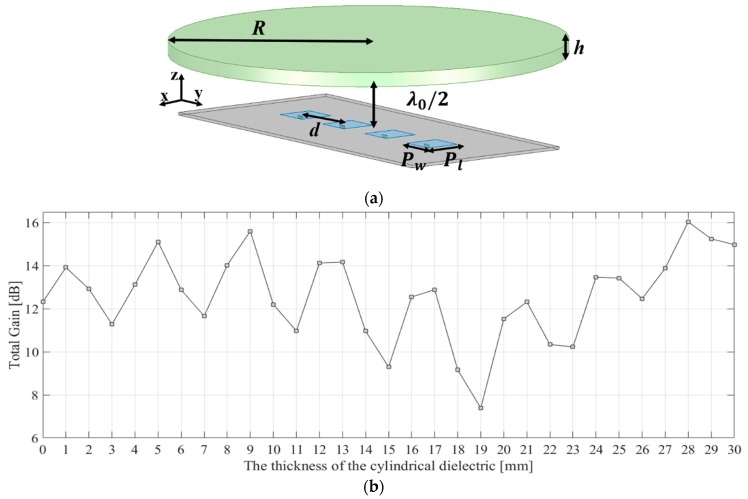
(**a**) Simulation model for the cylindrical dielectric medium fed by a 1 × 4 feed antenna array and (**b**) total gain of the cylindrical dielectric medium fed by the 1 × 4 antenna array as a function of the thickness of the medium.

**Figure 4 sensors-19-00538-f004:**
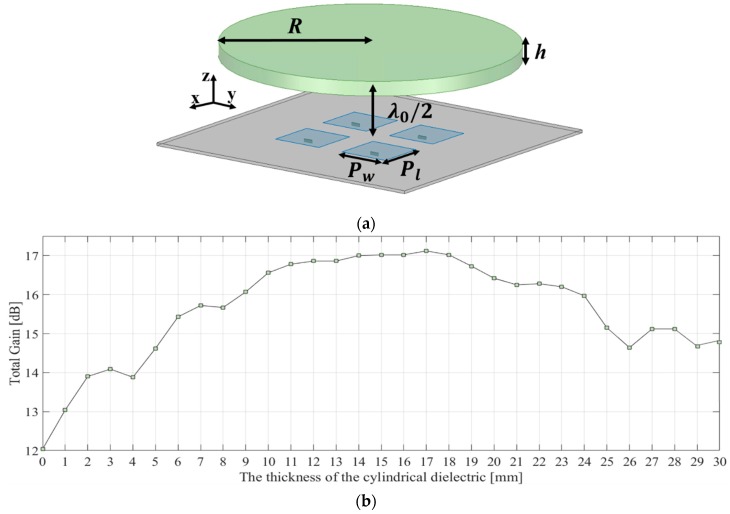
(**a**) Simulation model for the cylindrical dielectric medium fed by a 2 × 2 feed antenna array and (**b**) total gain of the cylindrical dielectric medium fed by the 2 × 2 antenna array as a function of the thickness of the medium.

**Figure 5 sensors-19-00538-f005:**
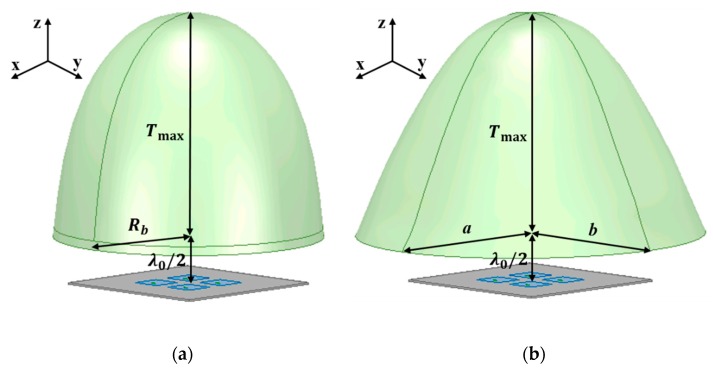
2 × 2 antenna array-fed dielectric lenses designed by (**a**) the PE method and (**b**) the HP method.

**Figure 6 sensors-19-00538-f006:**
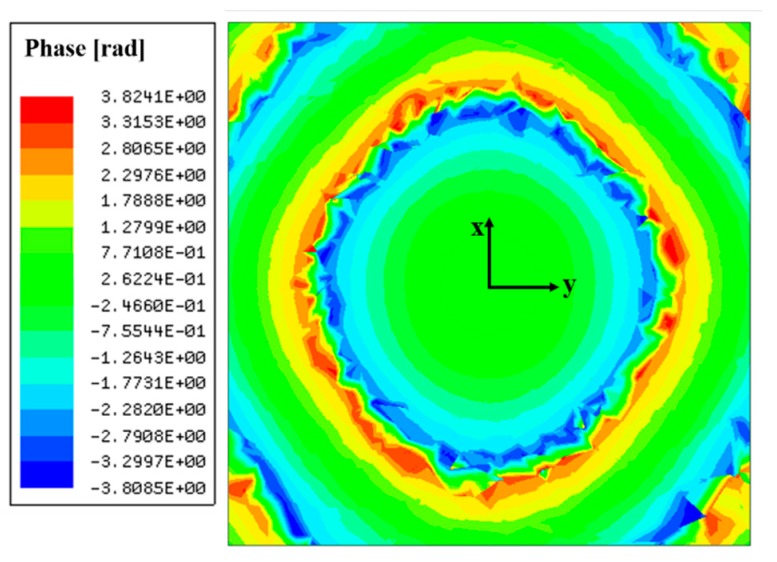
Captured phase distributions at a distance of *λ*_0_/2 away from the 2 × 2 antenna array.

**Figure 7 sensors-19-00538-f007:**
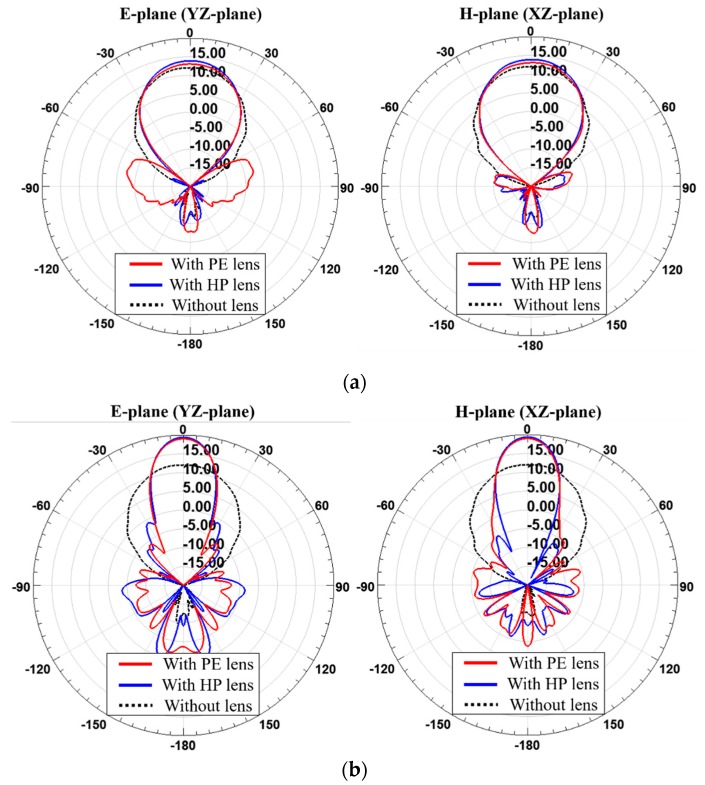
Simulated radiation patterns of the 2 × 2 antenna array with and without PE lens and HP lens at 28 GHz when (**a**) *T*_max_ is 4 mm and (**b**) *T*_max_ is 26 mm.

**Figure 8 sensors-19-00538-f008:**
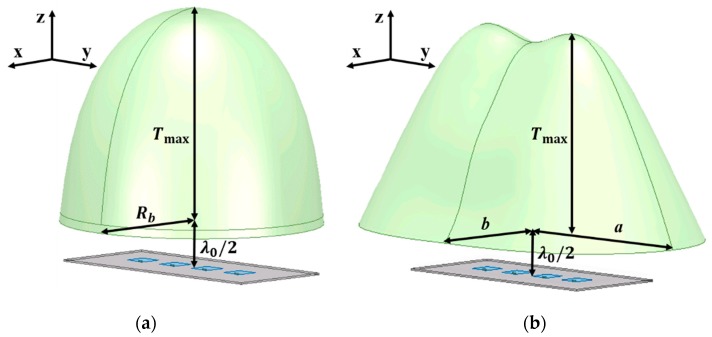
1 × 4 antenna array-fed lenses designed by (**a**) the PE method and (**b**) the HP method.

**Figure 9 sensors-19-00538-f009:**
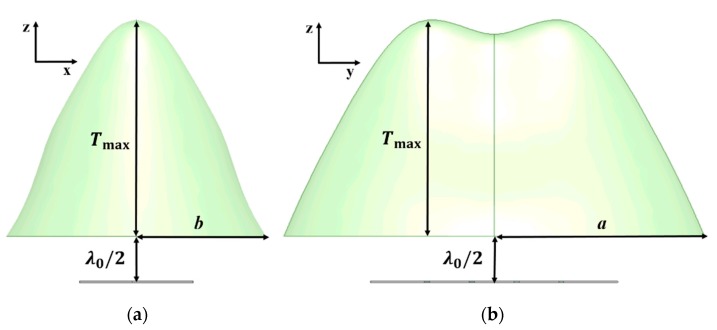
Cross sections of the 1 × 4 antenna array-fed HP lens in (**a**) the XZ plane and (**b**) the YZ plane.

**Figure 10 sensors-19-00538-f010:**
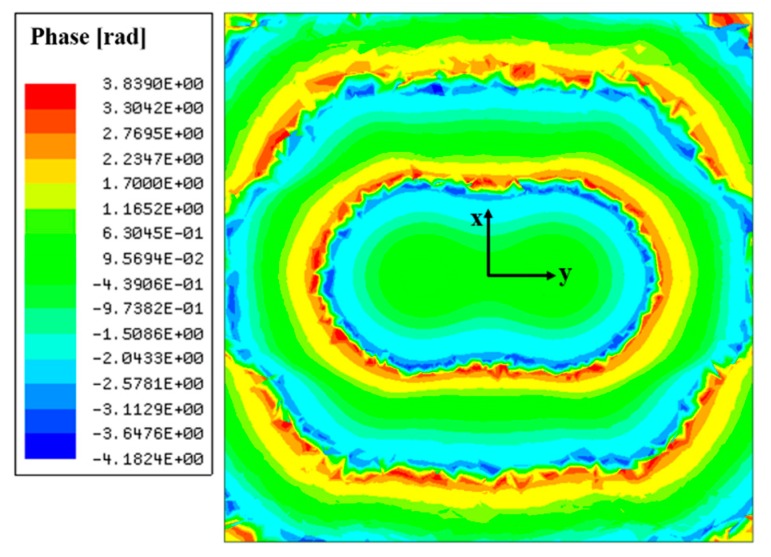
Captured phase distributions at a distance of *λ*_0_/2 away from the 1 × 4 antenna array.

**Figure 11 sensors-19-00538-f011:**
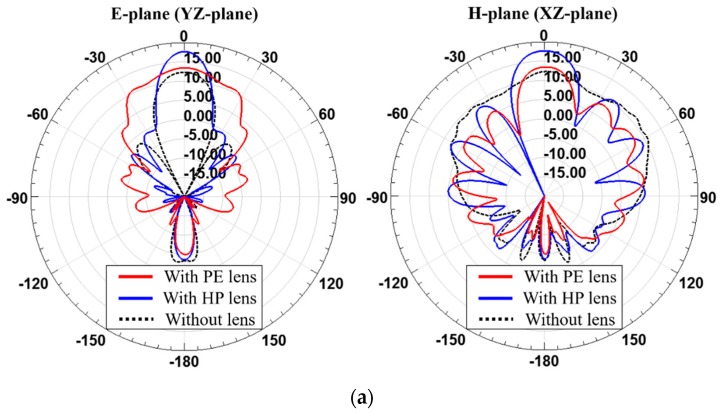
Simulated radiation pattern of the 1 × 4 antenna array with and without PE lens and HP lens at 28 GHz when (**a**) *T*_max_ is 19 mm and (**b**) *T*_max_ is 21 mm.

**Figure 12 sensors-19-00538-f012:**
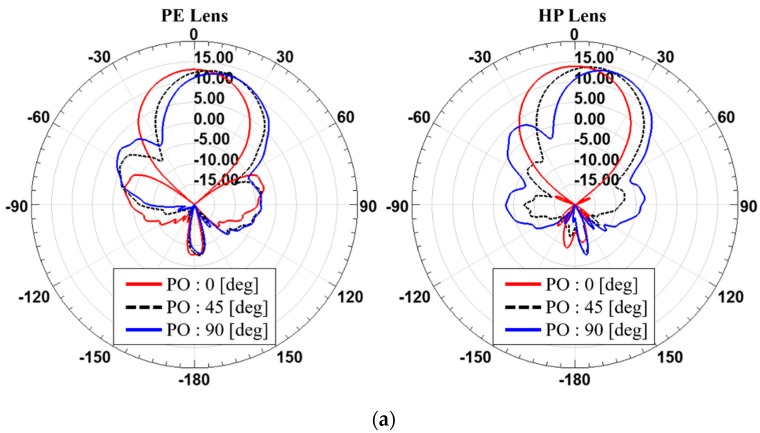
Comparison between with the PE lens and the HP lens in terms of beam steering when (**a**) *T*_max_ is 4 mm and (**b**) *T*_max_ is 26 mm for the 2 × 2 antenna array.

**Figure 13 sensors-19-00538-f013:**
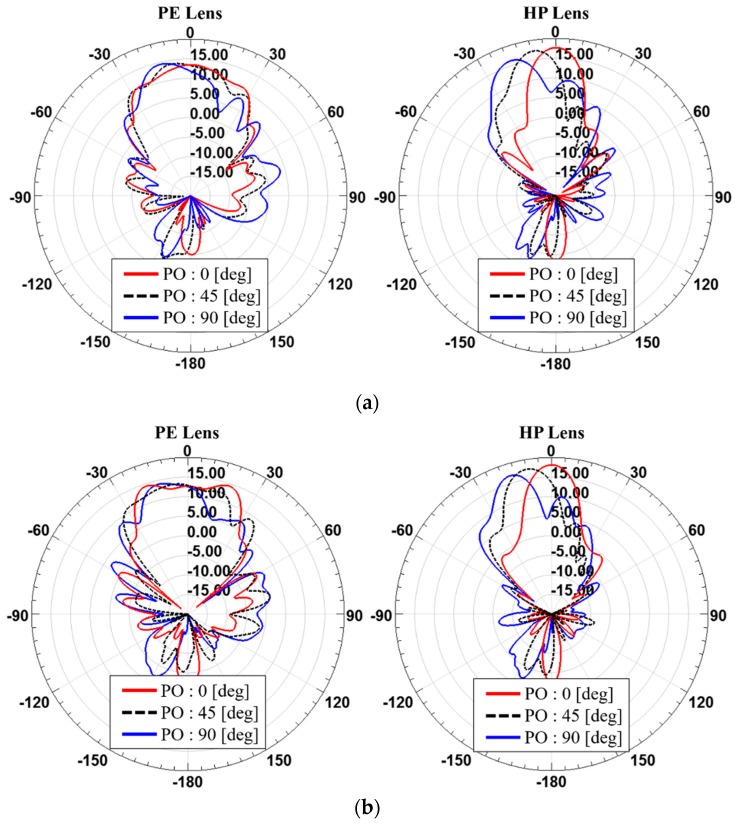
Comparison between with the PE lens and the HP lens in terms of beam steering when (**a**) *T*_max_ is 4 mm and (**b**) *T*_max_ is 26 mm for the 1 × 4 antenna array.

**Table 1 sensors-19-00538-t001:** Dimension parameters of the 2 × 2 antenna array-fed dielectric lens based on the PE method.

*T*_max_ (mm)	*R_b_* (mm)
4	4.6
26	16

**Table 2 sensors-19-00538-t002:** Dimension parameters of the 2 × 2 antenna array-fed dielectric lens based on the HP method.

*T*_max_ (mm)	*a* (mm)	*b* (mm)
4	8.6	7.7
26	21.3	19.5

**Table 3 sensors-19-00538-t003:** Simulated antenna gain with PE lens or HP lens according to *T*_max_.

*T*_max_ (mm)	PE Lens (dBi)	HP Lens (dBi)
4	13.07	13.90
26	19.02	19.40

**Table 4 sensors-19-00538-t004:** Dimension parameters of the 1 × 4 antenna array-fed dielectric lens based on the PE method.

*T*_max_ (mm)	*R_b_* (mm)
19	12.7
21	13.6

**Table 5 sensors-19-00538-t005:** Dimension parameters of the 1 × 4 antenna array-fed dielectric lens based on the HP method.

*T*_max_ (mm)	*a* (mm)	*b* (mm)
19	22	12
21	23	12.7

**Table 6 sensors-19-00538-t006:** Simulated antenna gain with the PE lens or the HP lens according to *T*_max_.

*T*_max_ (mm)	PE Lens (dBi)	HP Lens (dBi)
19	13.53	17.67
21	12.68	18.11

**Table 7 sensors-19-00538-t007:** Comparison of simulated gains between with the PE lens and the HP lens when *T*_max_ is 4 mm for the 2 × 2 antenna array.

Lens	PO (deg)	Tilted Angle (deg)	Gain (dBi)
	0	0	13.07
**PE**	45	10.53	13.07
	90	15.32	12.67
	0	0	13.90
**HP**	45	9.52	13.90
	90	16.43	13.75

**Table 8 sensors-19-00538-t008:** Comparison of simulated gains between with the PE lens and the HP lens when *T*_max_ is 26 mm for the 2 × 2 antenna array.

Lens	PO (deg)	Tilted Angle (deg)	Gain (dBi)
	0	0	19.02
**PE**	45	3.45	18.54
	90	8.14	17.64
	0	0	19.40
**HP**	45	4	18.96
	90	7.87	18.14

**Table 9 sensors-19-00538-t009:** Comparison of simulated gains between with the PE lens and the HP lens when *T*_max_ is 19 mm for the 1 × 4 antenna array.

Lens	PO (deg)	Tilted Angle (deg)	Gain (dBi)
	0	0	13.53
**PE**	45	−7.04	14.02
	90	−13	14.47
	0	0	17.67
**HP**	45	−8.70	17.24
	90	−18.31	16.19

**Table 10 sensors-19-00538-t010:** Comparison of simulated gains between with the PE lens and the HP lens when *T*_max_ is 21 mm for the 1 × 4 antenna array.

Lens	PO (deg)	Tilted Angle (deg)	Gain (dBi)
	0	0	12.68
**PE**	45	−5.94	13.33
	90	−13	13.98
	0	0	18.11
**HP**	45	−9.8	17.52
	90	−16.91	16.87

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
