# Peer review of "Practical Design Considerations for Compact Array-Fed Huygens’ Dielectric Lens Antennas"

_sensors, 2019, doi:10.3390/s19030538_

Reviewer 1 Report

In this manuscript, the some issues in the practical design of dielectric lens based on phase compensation are discussed. It is addressed that certain ranges of dielectric thickness values are not considered to exclude undesired resonant effects. Based on the Huygens’ principle, the phase distribution are calculated as an initial value for the design of dielectric lenses. Simulation results show that a gain enhancement of 5.34 dB can be achieved compared with conventional dielectric lenses. The method is really useful for practical design and the results sound interesting and solid. Thus this manuscript can be published after some minor issues are addressed.

Here are some minor comments:

(1) How do you get the phase distribution on the target plane? Is it a simulation result or numerical calculation?

(2) The distance between the lens and the array is lambda0/2 in this paper. Is it a fixed value or an optimized value?

(3) It can be seen from the simulation results, when the phase distribution of the array is similar to that of a point source, the difference between the compensation effects of HP and PE method is small. When the phase distribution is much different from that of a point source, the compensation effect of HP method is significant. The simulation results also verified the theoretical design. There is only one main concern that the area of the HP based lens is bigger than that of a PE based lens, for both 1 * 4 array and 2 * 2 array. So do you consider this point for the gain enhancement?

(4) In Fig. 5, for the phase distribution, there should be x- and y-axis showing the dimension. Also the text in the color bar is a little bit small. This case also occurs to Fig. 9.

(5) Some related works based on the phase compensation method can be referred to, including:

Yueyi Yuan, et al. Complementary transmissive ultra-thin meta-deflectors for broadband polarization-independent refractions in the microwave region. Photonics Research, 7(1), 80-88, 2019.

Author Response

Response to Reviewer 1 Comments

In this manuscript, the some issues in the practical design of dielectric lens based on phase compensation are discussed. It is addressed that certain ranges of dielectric thickness values are not considered to exclude undesired resonant effects. Based on the Huygens’ principle, the phase distribution are calculated as an initial value for the design of dielectric lenses. Simulation results show that a gain enhancement of 5.34 dB can be achieved compared with conventional dielectric lenses. The method is really useful for practical design and the results sound interesting and solid. Thus this manuscript can be published after some minor issues are addressed.

Response: Authors appreciate careful review of the paper by Reviewer 1 and the valuable comments.

- Authors have addressed all comments/suggestions from Reviewer 1 and made extensive amendments to the manuscript. Thanks.

- Authors have the revised manuscript checked by MDPI English editing service and the certificate of the English editing is attached as shown below.

Here are some minor comments:

Point 1: (1) How do you get the phase distribution on the target plane? Is it a simulation result or numerical calculation? 

Response 1: The phase distributions are acquired by a full-wave simulator, Ansys HFSS. Phase information of an e-field component corresponding to antenna polarization is captured along the lens aperture at the target distance away from the antenna array. This information is added in the revised manuscript.  

Point 2: The distance between the lens and the array is lambda0/2 in this paper. Is it a fixed value or an optimized value?

Response 2: The λ0/2 is chosen and fixed as a typically used distance to realize a transparent medium based on the fact that microwave impedance matching features are repeated every λ0/2. This information is added in the revised manuscript.  

Point 3: It can be seen from the simulation results, when the phase distribution of the array is similar to that of a point source, the difference between the compensation effects of HP and PE method is small. When the phase distribution is much different from that of a point source, the compensation effect of HP method is significant. The simulation results also verified the theoretical design. There is only one main concern that the area of the HP based lens is bigger than that of a PE based lens, for both 1 * 4 array and 2 * 2 array. So do you consider this point for the gain enhancement?

Response 3: Thanks for the useful comment. Yes, authors absolutely consider the point but we think that the current version of the manuscript doesn’t explain its reason explicitly. The following explanation in the next paragraph is added in the revised manuscript:

It should be noted that once F, Tmax, and n in Fig. 1 are chosen maximum available radius of the lens is decided by the total reflection condition at the top surface of the lens. In more detail, once a design point on the lens is chosen by η to decide the dielectric thickness at that point, F, Tmax, n, and η become known. Accordingly, other five variables r, γ, l, α, and s can be solved by five equations (1)-(5). As α goes over αc, reflections on the top surface of the lens goes into the total reflection condition and thus no solutions exist for the five variables related to the lateral size of the lens by Rb = r * sin(ηmax). Therefore, if one want to further increase the lateral size of the dielectric lens, the fixed values for F, Tmax, and n must be changed. For example, the increase of Tmax allows for the increase of the lateral dimension of the lens. Unlike the PE method assuming a point source, HP method that can consider multiple sources practically allows for design of the wider dielectric lens. This is why once F, Tmax, and n are fixed for fair comparisons, the lateral dimensions of the two different method-based lenses can be different, suggesting practical advantages of the HP method.    

- Fig. 1 is revised for better understanding of the above explanation.  

- (4) and (5) are added in the revised manuscript.

- The above paragraph is added in the revised manuscript.

Point 4: In Fig. 5, for the phase distribution, there should be x- and y-axis showing the dimension. Also the text in the color bar is a little bit small. This case also occurs to Fig. 9.

Response 4: Thanks for the observation. The figures (Fig. 6 and Fig. 10 in the revised manuscript) are improved.

Point 5: Some related works based on the phase compensation method can be referred to, including:

Yueyi Yuan, et al. Complementary transmissive ultra-thin meta-deflectors for broadband polarization-independent refractions in the microwave region. Photonics Research, 7(1), 80-88, 2019.

Response 5: Thanks for the suggestion. The literature is added in the reference list of the revised manuscript.

Reviewer 2 Report

The authors report a new method to enhance antenna gain with the dielectric lens. Unlike to conventional method, which is called the parabolic equation (PE) method, the proposed method, namely Huygens’ principle (HP) method, can effectively enhance antenna gain even with an asymmetric antenna array configuration, such as 1 by 4 antenna array configuration. Although the manuscript has some interesting points, the experimental part is fully absent, that is, the proposed method has not been confirmed. Accordingly, I strongly suggest the authors to include experimental results along with the experimental method. Additionally, there are some issues that must be addressed and clarified in the manuscript to be considered for publication in Sensors. Please find my detailed remarks below:

1.       The authors stated in page 3 line 73 “It should be noted that most antenna applications use antenna arrays for beam steering capability. Therefore, the assumption of a single point source in the above PE method limits achieving optimal lens gain for the feed antenna arrays, multiple and arbitrarily arranged radiating sources.” However, there are no studies or discussions on beamforming performance of antenna arrays with PE and HP dielectric lens throughout the manuscript. I suggest the authors to provide the effectiveness of HP dielectric lens on beam steering performance of antenna arrays and prove that the dielectric lens antenna array with HP outperforms the dielectric lens antenna array with PE method.

 2.       In page 5, the authors simulated two different antenna array configurations with the cylindrical dielectric medium to compare the effect of the dielectric medium on antenna array performance with symmetric and asymmetric array configuration. However, the authors only provided the thickness of the dielectric medium dependent gain of the 1 by 4 antenna array. It will be informative if authors provide the thickness dependent gain of the 2 by 2 antenna array.

 3.       In continuation of the preceding comment, the authors chose Tmax of 19 and 21 mm for the 1 by 4 antenna array and explained reasons for the parameter choice. However, in the case of the 2 by 2 antenna array, the authors did not provide any reasons for choosing values (4 mm and 26 mm) of Tmax. Please explain the reasons.

 4.       In page 6, the authors compare the antenna performance of 2 by 2 antenna array with PE lens and HP lens. In the case of the antenna array with PE lens, Rb was set to be 4.6 mm. Is the value of Rb optimized? Since the gain of 2 by 2 antenna array with PE lens with Rb of 4.6 mm (13.07 dBi) was degraded as compared to the gain of 2 by 2 antenna array with the cylindrical dielectric medium (13.88 dBi). How the authors chose the value of 4.6 mm for Rb?

 5.       In page 7, the authors provide the simulated radiation patterns of 2 by 2 antenna array with PE lens and HP lens. It will be more informative if the authors provide the radiation patterns without a dielectric lens.

 6.       Similar to comment #4, please explain how the authors chose values of Rb for PE lens.

7.    Similar to comment #5, it will be informative if the authors provide the radiation patterns without a dielectric lens in Fig. 10.

Author Response

Response to Reviewer 2 Comments

The authors report a new method to enhance antenna gain with the dielectric lens. Unlike to conventional method, which is called the parabolic equation (PE) method, the proposed method, namely Huygens’ principle (HP) method, can effectively enhance antenna gain even with an asymmetric antenna array configuration, such as 1 by 4 antenna array configuration. Although the manuscript has some interesting points, the experimental part is fully absent, that is, the proposed method has not been confirmed. Accordingly, I strongly suggest the authors to include experimental results along with the experimental method. Additionally, there are some issues that must be addressed and clarified in the manuscript to be considered for publication in Sensors. Please find my detailed remarks below:

Authors appreciate careful review of the paper by Reviewer 2 and the valuable comments.

- Authors have addressed all comments/suggestions from Reviewer 2 and made extensive amendments to the manuscript. Regarding Reviewer 2’s suggestion of including experimental results, authors absolutely agree with its solid values but authors have figured out that the fabrication of the proposed dielectric geometry in reliably high resolution smoothness is one of the challenging and evolving research themes in 3D/4D printing technology. Authors hope that in near-future publications we can show a novel 3D/4D printing methodologies to reduce the errors between simulated and measured results for such dielectric lenses. However, authors strongly believe that numerous computational case studies and corresponding consistent analysis in the revised manuscript are sufficient to validate the proposed ideas. This discussion is added in the conclusion of the revised manuscript as future works. Thanks.  

- Authors have the revised manuscript checked by MDPI English editing service and the certificate of the English editing is attached as shown below.

 Point 1: The authors stated in page 3 line 73 “It should be noted that most antenna applications use antenna arrays for beam steering capability. Therefore, the assumption of a single point source in the above PE method limits achieving optimal lens gain for the feed antenna arrays, multiple and arbitrarily arranged radiating sources.” However, there are no studies or discussions on beamforming performance of antenna arrays with PE and HP dielectric lens throughout the manuscript. I suggest the authors to provide the effectiveness of HP dielectric lens on beam steering performance of antenna arrays and prove that the dielectric lens antenna array with HP outperforms the dielectric lens antenna array with PE method.

Response 1: Thanks for the useful suggestion. The steered radiation patterns at the phase offsets of 0, 45, and 90 degrees for PE and HP lenses with the different array arrangement such as 2 by 2 and 1 by 4 and the different thicknesses of the dielectric lens are added in the revised manuscript. Also, the corresponding explanation is added in the manuscript.

It is found that for 2 by 2 antenna array such a symmetrical arrangement both of PE and HP lenses have similar beam steering features. However, for 1 by 4 antenna array HP lenses shows higher gain than PE lenses at all of the phase offsets. It is observed that PE lens has higher gains at steer angles, not at θ = 0 degree. This suggests that the PE method using theoretically ideal assumption of a point source is not relevant for practical dielectric lens antennas that must consider many different arrangement cases for the feed antenna array. 

- New figures (Fig. 12 and 13 in the revised manuscript) are added.

- New tables (Table 7, 8, 9, and 10 in the revised manuscript) are added.

- The above explanation is added in the revised manuscript.

Point 2: In page 5, the authors simulated two different antenna array configurations with the cylindrical dielectric medium to compare the effect of the dielectric medium on antenna array performance with symmetric and asymmetric array configuration. However, the authors only provided the thickness of the dielectric medium dependent gain of the 1 by 4 antenna array. It will be informative if authors provide the thickness dependent gain of the 2 by 2 antenna array.

Response 2: Thanks for the useful suggestion. A figure showing total gain of the cylindrical dielectric medium fed by the 1 by 4 antenna array as a function of the thickness of the medium is added in the revised manuscript.

It should be noted that the gain variations for 2 by 2 antenna array have different shapes compared with the gain variations for 1 by 4 antenna array due to the superposition of electromagnetic waves radiated by differently arranged multiple sources. 

- A new figure (Fig. 4 in the revised manuscript) is added.

- The above explanation is added in the revised manuscript.

Point 3: In continuation of the preceding comment, the authors chose Tmax of 19 and 21 mm for the 1 by 4 antenna array and explained reasons for the parameter choice. However, in the case of the 2 by 2 antenna array, the authors did not provide any reasons for choosing values (4 mm and 26 mm) of Tmax. Please explain the reasons.

Response 3: Sorry for the missing information. 

The 4 mm and 26 mm are chosen for total gain of the cylindrical dielectric-medium combined antenna array to be similar to the gain of the antenna array without any dielectric medium and thus gain enhancement effectiveness by phase compensation of the lens can be realized. The reason why 1 or 2 mm is not selected is that too small Tmax limits maximum available lateral dimension (Rb) of the lens due to total reflection condition on the top surface of the dielectric lens as stated in Section 2.1.

- The above explanation is added in the revised manuscript.

Point 4: In page 6, the authors compare the antenna performance of 2 by 2 antenna array with PE lens and HP lens. In the case of the antenna array with PE lens, Rb was set to be 4.6 mm. Is the value of Rb optimized? Since the gain of 2 by 2 antenna array with PE lens with Rb of 4.6 mm (13.07 dBi) was degraded as compared to the gain of 2 by 2 antenna array with the cylindrical dielectric medium (13.88 dBi). How the authors chose the value of 4.6 mm for Rb?

Response 4: Thanks for the useful comment. 

In Fig. 1, Rb is given by r * sin(ηmax). In more detail, total reflection condition on the top surface of the lens, α = αc, gives r and ηmax by solving (1)-(5). It should be noted that as α goes over αc, reflections on the top surface of the lens goes into the total reflection condition and thus no solutions exist for Rb. This limits maximum available lateral dimension of the lens for the fixed F, Tmax, and n. Through the aforementioned design procedure, when F, Tmax, and n are 5.3 mm, 4 mm, and 1.44, Rb is decided to be 4.6 mm.

- Fig. 1 is revised for better understanding of the above explanation.  

- (4) and (5) are added in the revised manuscript.

- The above paragraph is added in the revised manuscript.

Point 5: In page 7, the authors provide the simulated radiation patterns of 2 by 2 antenna array with PE lens and HP lens. It will be more informative if the authors provide the radiation patterns without a dielectric lens.

Response 5: Thanks for the suggestion. The suggested radiation patterns are added in Fig. 7 of the revised paper.

Point 6: Similar to comment #4, please explain how the authors chose values of Rb for PE lens.

Response 6: Again, thanks for the useful comment. Based on the design procedure explained in Response 4, when the values of Tmax are 4 and 26 mm for 2 by 2 antenna array, and 19 and 21 mm for 1 by 4 antenna array the values of Rb are decided to be 4.6, 16, 12.7, and 13.6 mm under the condition of the fixed F and n.   

Point 7: Similar to comment #5, it will be informative if the authors provide the radiation patterns without a dielectric lens in Fig. 10.

Response 7: Thanks for the suggestion. The suggested radiation patterns are added in Fig. 11 of the revised paper.

Round  2

Reviewer 2 Report

Thank you for the efforts made by the authors to revise the manuscript. The authors have responded to all of my earlier queries satisfactorily. I have no further questions or comments. Therefore, I suggest accepting the manuscript in current form.